# Four years (2011-2015) of Total Gaseous Mercury Measurements from the Cape Verde Atmospheric Observatory

Katie A Read[1], Luis M Neves[2], Lucy J Carpenter[1], Alastair C Lewis[1], Zoe L Fleming[3], and John Kentisbeer[4]

[1]National Centre for Atmospheric Science (NCAS), Department of Chemistry, University of York, York, YO10 5DD, UK
[2]Instituto Naçional de Meteorologia Geofisica (INMG), Delegãço de São Vicente, Monte, CP15, Mindelo, Rep of Cape Verde
[3]National Centre for Atmospheric Science (NCAS), University of Leicester, Leicester, LE1 7RH, UK
[4]Centre for Ecology and Hydrology (CEH), Bush Estate, Penicuik, Midlothian, EH26 0QB, UK

*Correspondence to:* Katie A. Read (katie.read@york.ac.uk)

**Abstract.** Mercury is a chemical with widespread anthropogenic emissions that is known to be highly toxic to humans, ecosystems and wildlife. Global anthropogenic emissions are around 20% higher than natural emissions and the amount of mercury released into the atmosphere has increased since the industrial revolution. In 2005 the European Union and United States adopted measures to reduce mercury use, in part to offset the impacts of increasing emissions in industrialising countries. The changing regional emissions of mercury have impacts on a range of spatial scales. Here we report four years (Dec 2011 – Dec 2015) of Total Gaseous Mercury (TGM) measurements at the Cape Verde Observatory (CVO), a global WMO-GAW station located in the sub-tropical remote marine boundary layer. Observed total gaseous mercury concentrations were between 1.03 and 1.33 ng m$^{-3}$ (10$^{th}$, 90$^{th}$ percentiles), close to expectations based on previous interhemispheric gradient measurements. We observe a decreasing trend in TGM (-0.05 ± 0.04 ng m$^{-3}$ yr$^{-1}$, -4.2% ± 3.3% yr$^{-1}$) over the four years consistent with the reported decrease of mercury concentrations in North Atlantic surface waters and reductions in anthropogenic emissions. The decrease was more visible in the summer (Jul-Sep) than in the winter (Dec-Feb), when measurements were impacted by air from the African continent and Sahara/Sahel regions. African air masses were also associated with the highest and most variable TGM concentrations. We suggest that the less pronounced downward trend inclination in African air may be attributed to poorly controlled anthropogenic sources such as artisanal and small-scale gold mining (ASGM) in West Africa.

## 1 Introduction

Mercury is present in the atmosphere in three main forms; gaseous elemental mercury Hg$^0$, which is the most common form in the gas phase, oxidized mercury Hg$^{II}$ (GOM or RGM), and Hg-bound to particulate matter (PBM). Total Gaseous Mercury (TGM) is the combined measurement of Hg$^0$ (or Gaseous Elemental Mercury (GEM)) + RGM, with Hg$^0$ typically contributing around 90-99% of the total Hg or TGM.

Anthropogenic sources of mercury account for around 30% of the total amount and include emissions from coal burning, mining, cement production, oil refining and waste incineration. One third of the anthropogenic emissions are thought to come from deliberate biomass burning with Africa as the single largest continental source; therefore in this region there could be an influence from Sahel African biomass burning during the months of November through to February (Roberts et al, 2009, De Simone et al.,2015). Hg$^0$ reacts slowly with atmospheric oxidants with a global lifetime of around 6-8 months (Selin et al., 2007;Holmes et al., 2010), and so can be transported to remote regions. When oxidized to less volatile Hg$^{II}$, it can be deposited either through wet deposition processes (precipitation-scavenging) or by surface uptake (Gustin et al., 2012;Schroeder and Munthe, 1998;Sather et al., 2013;Wright et al., 2014). Hg$^0$ also undergoes slow dry deposition through air-surface exchange with both terrestrial and aquatic surfaces (Zhang et al., 2009;Wang et al., 2016). Once deposited, transformation to highly toxic species such as the neurotoxic methylmercury allows bioaccumulation in food chains and poses a health risk to humans and a damaging effect to ecosystems (US EPA, 1997). Previously deposited mercury can also be reduced back to Hg$^0$ through the natural weathering of

mercury-containing rocks, geothermal activity, or from volcanic activity, and then re-emitted back to the atmosphere (Smith et al., 2008)(Qureshi., 2012).

Reactions of $Hg^0$ to $Hg^{II}$ with the hydroxyl radical (OH) and ozone ($O_3$) were historically accepted as the dominant photochemical oxidation mechanisms (Bergan and Rodhe, 2001;Lin et al., 2006;Seigneur et al., 2006;Selin et al., 2007;Pongprueksa et al., 2008). Recent work has suggested that there may be significant other oxidants such as atomic halogens (Holmes et al., 2010;Wang et al., 2014) and more complex two-step oxidation schemes, which include further reactions with $NO_2$ and $HO_2$, however the kinetics are highly uncertain (Goodsite et al., 2004) (Goodsite et al., 2004;Holmes et al., 2010). Heterogeneous oxidation in clouds may also contribute but is not experimentally proven (Ariya et al., 2009;Calvert and Lindberg, 2005).

Strode et al. (2007) estimated that 36% of all mercury emissions in the northern hemisphere come from the ocean both through primary emission (ocean upwelling and mercury-containing rocks) and from re-emission of previously deposited mercury (as $Hg^{II}$), but this increases to 55% as you move into the southern hemisphere (Strode et al., 2007). The major anthropogenic source affecting the remote marine boundary layer is likely to be long-range transport of $Hg^0$ from combustion (smelting, waste incineration, chemical plants) rather than from $Hg^{II}$, which is more likely to deposit regionally due to its relatively short lifetime of 4.8 hours (Zhang et al., 2012). Other industrial sources for Hg include artisanal and small-scale gold mining (ASGM), which are known to occur in West Africa (Telmer and Velga, 2009; UNEP, 2013) and will likely regionally influence the measurements described here. For the 2013 UNEP global assessment, ASGM emission data were compiled from field and industry reports but with an uncertainty of ca. ± 43% due to the multitude and varying nature of ASGM sites. In recent years, global emissions from ASGM, and in particular the proportion of global emissions attributed to South America and Sub-Saharan Africa, appear to be increasing; however this assumption may be due to improved reporting (Muntean et al., 2014). The majority of global anthropogenic emissions of Hg to the atmosphere in 2010 are associated with ASGM (37%), with one third thought to be from sub-Saharan Africa (UNEP, 2013).

A community strategy developed by the EU was adopted in 2005 and listed 20 actions to reduce mercury emissions, cut mercury supply and demand, and to protect people against exposure. This strategy had a strong focus on the need to take a global approach and included actions relating to multilateral negotiations for the conclusion of a legally binding convention on mercury (http://ec.europa.eu/environment/chemicals/mercury/strategy_en.htm). The UNEP Global Mercury partnership led by the US Environmental Protection Agency took a similar approach (http://www.unep.org/chemicalsandwaste/Metals/GlobalMercuryPartnership/tabid/1253/Default.aspx) and these initiatives formed the basis of the Minamata Convention on Mercury, which was agreed in 2013 and is a global treaty to protect human health and the environment from the adverse effects of mercury (http://www.mercuryconvention.org/).

It has been a source of contradiction that in the northern hemisphere, while both measured atmospheric Hg concentrations and wet deposition fluxes have been decreasing since 1990 (Soerensen et al., 2012) and 1996-2013 (Slemr et al., 2011;Weigelt et al., 2015); global Hg emissions during this period were calculated to be increasing (Pacyna et al., 2010;Streets et al., 2011). Very recently, however, Zhang et al., (2016), using a revised inventory and the global model GEOS-CHEM, have shown that global Hg emissions may also be decreasing. They suggest that a large discrepancy in the emissions data was from locally deposited mercury close to coal-fired utilities. It is thought that this source has declined more rapidly than was previously predicted due to shifts in mercury speciation from air pollution control technology targeted at $SO_2$ and $NO_x$ (Zhang et al., 2016). Flue gas desulfurization (FGD) - which controls $SO_2$ emissions - washes out $Hg^{II}$, whilst selective catalytic reduction (SCR) to control $NO_x$ emissions also oxidises $Hg^0$ to $Hg^{II}$. These effects of FGD, in addition to the recent phase-out of Hg from commercial products (UNEP Minamata Convention on Mercury) and lower global estimates from small-scale gold mining, serve to explain the globally decreasing atmospheric concentrations in the model. Zhang et al. (2016) also found that the larger emission decreases observed in North America and Europe globally offset the increases from other major polluted regions e.g. from coal-fired utilities in East Asia (Pacyna et al., 2010;Pirrone et al., 2013).

Using data from ship cruises, Soerensen et al. (2012) observed a significant decreasing trend of atmospheric mercury concentrations over the North Atlantic of -0.046 ng m$^{-3}$ yr$^{-1}$ (-2.5% yr$^{-1}$), with smaller trends at more southern latitudes (Soerensen et al., 2012). They suggest that this decline is due

to decreasing oceanic evasion driven by declining subsurface water Hg$^0$ concentrations (-5.7% yr$^{-1}$
since 1999, (Mason et al., 2012)).
Here we report four years (Dec 2011 – Dec 2015) of TGM measurements at the Cape Verde
Observatory (CVO), a clean marine background station located in the subtropical Atlantic.  The
measurements presented here are part of the EU Global Mercury Observation System (GMOS)
network.  The GMOS network of sites was established in 2011 with the aim of addressing known gaps
in the spatial and temporal measurement of mercury, as well as improving knowledge of Hg speciation.
The data is being used to validate regional and global scale atmospheric Hg models in order to improve
understanding of global Hg transport, deposition and re-emission as well as providing a contribution to
future international policy development and implementation (www.gmos.eu).
**2  Experimental**
The CVO was established in 2006 as a multilateral project between the UK, Germany and Republic of
Cape Verde.   Long-term atmospheric measurements include reactive trace gases including ozone,
carbon monoxide, nitrogen oxides and volatile organic compounds (National Centre for Atmospheric
Sciences (NCAS), University of York, UK), long lived greenhouse gases (Max-Planck Institute (MPI),
Jena, Germany), and physical and chemical characterisation of aerosol (Leibniz Institute for
Tropospheric Research (TROPOS), Leipzig, Germany). Details of the measurements and
characteristics of the station can be found in Carpenter et al., (2010).
The CVO is positioned on the northeast side of Sao Vicente (16.85°N, 24.87°W), one of ten islands in
the Cape Verde archipelago (Fig. 1).  The island is of volcanic origin and the CVO is situated 50m
from the coastline.  The climate is warm (mean annual air temperature is 24.0°C ± 2.0°C) and dry with
extremely low annual rainfall (<200 mm), which occurs mostly during the rainy season of July-
November.  The site receives air masses from the northeasterly trade winds for 95% of the time, which
have travelled typically for five days over the ocean.  Research flights carried out over the CVO in
summer 2007 established that the boundary layer is well mixed (Read et al., 2008).  There is no coastal
shelf to the island at this location point, and the CVO conditions are considered to be representative of
the North Atlantic open ocean boundary layer.  Radiosonde and ceilometer data show there is no
diurnal pattern evident in boundary layer heights, which suggests no systematic difference between
33 day-time and night-time entrainment rates (Carpenter et al., 2010).

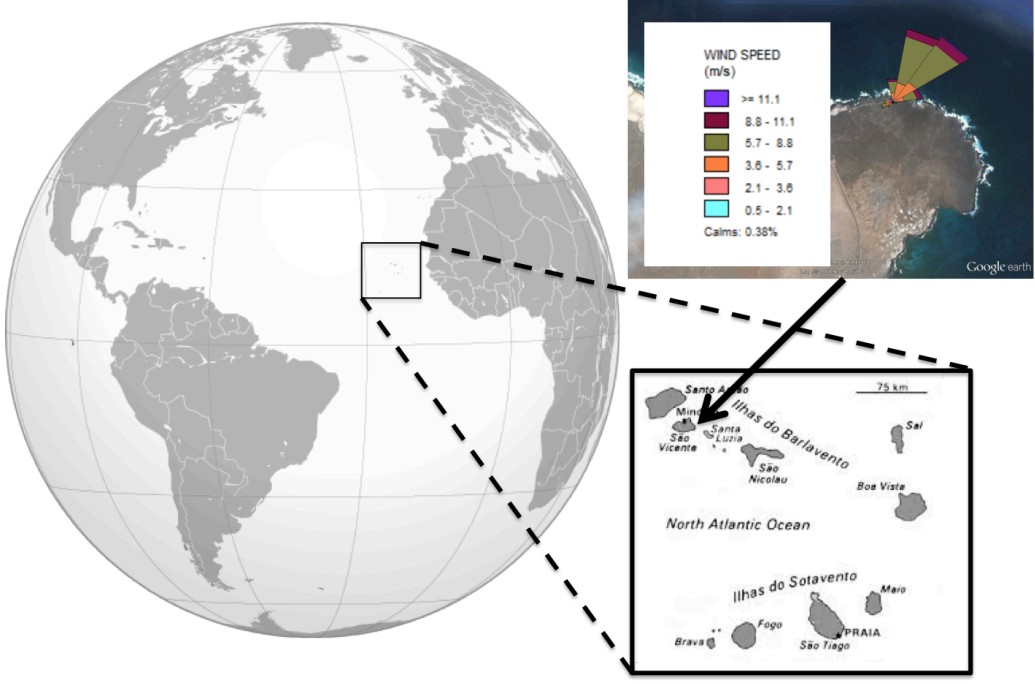

**Figure 1: Cape Verde site location.  Top right, image from Google earth: V7.1.5.1557 (6$^{th}$ July 2016). São Vicente, Cape Verde, 16°51'59.60''N, 24° 52'03.60''W, Eye altitude 2.70 km.   Wind rose for the measurement period is coloured by wind speed.**

Air is sampled from the main laboratory glass manifold (10 m height of inlet, 2" diameter, residence time 4 seconds) and then through a 2 m length of ¼" Teflon tubing and a particulate filter which is changed every two months. The entire inlet is heated. A TEKRAN 2537B analyser (Tekran Inc., Toronto, Canada) was used for the TGM measurements and is described in detail elsewhere (Steffen et al., 2014) and so only a brief summary is presented here. The analytical principle collects the TGM onto gold traps with subsequent thermal desorption and detection by atomic fluorescence spectroscopy ($\lambda$= 253.7 nm, (Bloom and Fitzgerald, 1988)). It is however likely that the measurement at this site is of GEM rather than TGM since RGM is lost very easily to any salt deposits in the inlet lines and filters. Samples of 5 L volume are obtained every 5 minutes (1 L min$^{-1}$ flow rate) with a detection limit of around 0.1 ng m$^{-3}$, using a dual trap set-up. Concentrations in ng m$^{-3}$ are reported at a standard pressure of 1013 hPa and a standard temperature of 273.14K. Calibrations are performed every 72 hours using an internal mercury permeation source which injects a known amount of Hg$^0$ into mercury-free zero air (using a TEKRAN Zero Air filter, part no: 90-25360-00). The calibration consists of a zero and a span on each channel. The effective span was 19.08 ng m$^{-3}$ for a sample volume of 5 L. The permeation rate was externally validated using manual injections of saturated mercury vapour taken from a Tekran 2505 mercury vapour calibration unit and after 5 years found to be within ~3.6% of the instrumental set-point. The detection limit of the instrument was 0.1 ng m$^{-3}$.

Instruments to make trace gas and meteorological measurements are provided by the Atmospheric Measurement Facility (AMF), which is part of the National Centre for Atmospheric Science (NCAS). Ozone measurements were made using a UV photometric analyser (Thermo Electron Corporation). The instrument had a detection limit of 0.05 ppb and a precision of <1 ppbV. Carbon monoxide data presented here was measured using a Vacuum UV fluorescence technique (Aerolaser 5001). It was sensitive to 1 ppbV and linear up to 100 ppm. The accuracy of the measurements was <2ppbV. In 2012 the Global Atmospheric Watch (GAW) audited these measurements and the report can be found at http://www.wmo.int/pages/prog/arep/gaw/documents/CVO_2012.pdf. Nitrogen oxide measurements were made using a low detection, high accuracy, (sensitivity 0.3 and 0.35 pptV and accuracy of 5.5% and 5.9% for NO and NO$_2$ respectively) chemiluminescence analyser (AQD Inc.). The meteorological measurements presented here were made using a Campbell Scientific Automatic Weather station. For more information about the present NCAS instrumentation at the CVAO refer to https://www.ncas.ac.uk/index.php/en/the-facility-amf/291-amf-main-category/cvao/cvao-amf-instrumnets/1557-cvao-amf-instrumnets.

Four years of data are presented here obtained between 5[th] December 2011 and 5[th] December 2015. In calculating annual statistics, we have used data from 1 Dec – 30 Nov. The data was quality controlled using the central GMOS-Data Quality Management (G-DQM) system (Cinnirella et al., 2014;D'Amore et al., 2015). The G-DQM allows harmonization of data across the network and is able to acquire and process data in near real time allowing immediate diagnosis of issues. It was developed using harmonized Standard Operating Procedures, which had been established over many years by European and Canadian monitoring networks, together with recent literature (Brown et al., 2010;Gay et al., 2013;Steffen et al., 2012). An additional filter has been applied to the data presented here to exclude periods when the relative humidity was higher than 90%, as the data was prone to increased uncertainties due to water condensing in the instrument. Instrument issues led to some significant data gaps; a lamp failure caused major data gaps between July-August 2012 and May-June 2014, whilst a pump failure caused downtime between October 2012-January 2013.

## 3 Results and discussion

### 3.1 Statistics and seasonal cycles

The mean TGM concentration over 2011-2015 was 1.191 ± 0.128 ng m$^{-3}$ and the four-year time-series is shown in Fig. 2. Sprovieri et al. (2016) showed that the CVO measurements (site referred to as CAL rather than CVO) fit well within the north-south gradient of TGM data. Other sites of reference, which receive background air similar in origin to CVO, include Mace Head, Ireland, Nieuw Nickerie, Suriname, and Cape Point, South Africa (Table 1). The remoteness of the CVO is reflected in the small variability of the TGM measurements, compared to other sites.

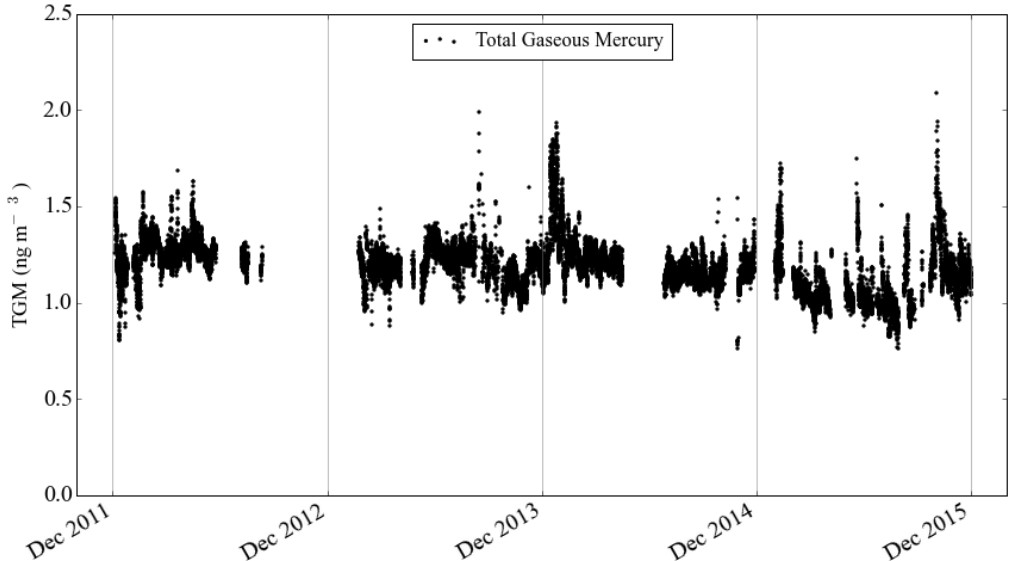

**Figure 2: Time-series (December 2011-December 2015) of TGM data measured at the Cape Verde Observatory.**

The data shown in Table 1 illustrate the dominating effect of emissions from the northern hemisphere compared to the southern hemisphere, with Mace Head (53°20'N, 9°54'W) TGM concentrations averaging 7-9% higher than those observed at Cape Point. The site at Niew Nickerie experiences 10% higher concentrations in the air arriving from the north compared to the south (Muller et al., 2012) and is additionally impacted by emissions from biomass burning and gold mining from South America (Sprovieri et al., 2010). Comparisons to ship-borne field campaigns in the Atlantic made between 1977 and 2001 (Sprovieri et al., 2010) show that the data from CVO are more comparable with southern Atlantic conditions than the northern Atlantic.

| Site (Latitude, Longitude) | Mean ± standard deviation for 2013 (ng m$^{-3}$) | Mean ± standard deviation for 2014 (ng m$^{-3}$) |
|---|---|---|
| Mace Head, Ireland (53°20'N, 9°54'W) | 1.46 ± 0.17 | 1.41 ± 0.14 |
| Calhau, Rep of Cape Verde (16°51'N, 24°52'W) | 1.22 ± 0.14 | 1.20 ± 0.09 |
| Nieuw Nickerie, Suriname (5°56'N, 56° 59'W) | 1.13 ± 0.42 | 1.28 ± 0.46 |
| Cape Point, South Africa (33° 56'S, 18°28'E) | 1.03 ± 0.11 | 1.09 ± 0.12 |

**Table 1. Average TGM concentrations and standard deviation statistics from comparable sites in 2013 and 2014. Data from Sprovieri et al., (2016).**

The CVO TGM monthly mean data shows a weak seasonal cycle (1.289 ± 0.134 ng m$^{-3}$ December maximum, 1.130 ± 0.128 ng m$^{-3}$ June minimum, Fig. 3) with generally higher concentrations in winter and lower in summer. This cycle is similar, both in shape and magnitude, to that observed within sub-tropical maritime air masses at Mace Head, which is shallower than for other air masses (Weigelt et al., 2015) and generally not so defined as that of other remote sites in the Northern Hemisphere (Temme et al., 2007;Holmes et al., 2010). Selin et al., (2007) show that the mean seasonal amplitude of 12 northern mid-latitude sites between the maximum in January (winter) and minimum August (summer) is 0.19 ng m$^{-3}$, compared to the CVO amplitude of 0.14 ng m$^{-3}$ (December-June).
A smaller seasonal cycle may mean that $O_3$ plays a more dominant role in the oxidation of $Hg^0$ compared to other oxidants such as OH (Temme et al., 2007;Selin et al., 2007;Holmes et al., 2010). The equatorial nature of the CVO site means that solar irradiance and water vapour are high year-round (Carpenter et al., 2010;Whalley et al., 2010). This may lead to a less pronounced change in oxidation capacity between summer and winter, when compared to sites at higher latitudes.

An influence of air masses from the southern hemisphere without any pronounced seasonal variation (Slemr et al., 2015) may be another reason for the smaller amplitude in the seasonal cycle at the CVO. However, air mass back trajectory analyses show that the CVO receives very little air representative of the southern hemisphere (~1.3% of all data, Fig. 1, Supplementary Information). Further, the highest frequency of southerly air masses arriving at the CVO occurs during August and September, which would serve to increase the mercury seasonal cycle amplitude rather than reduce it.

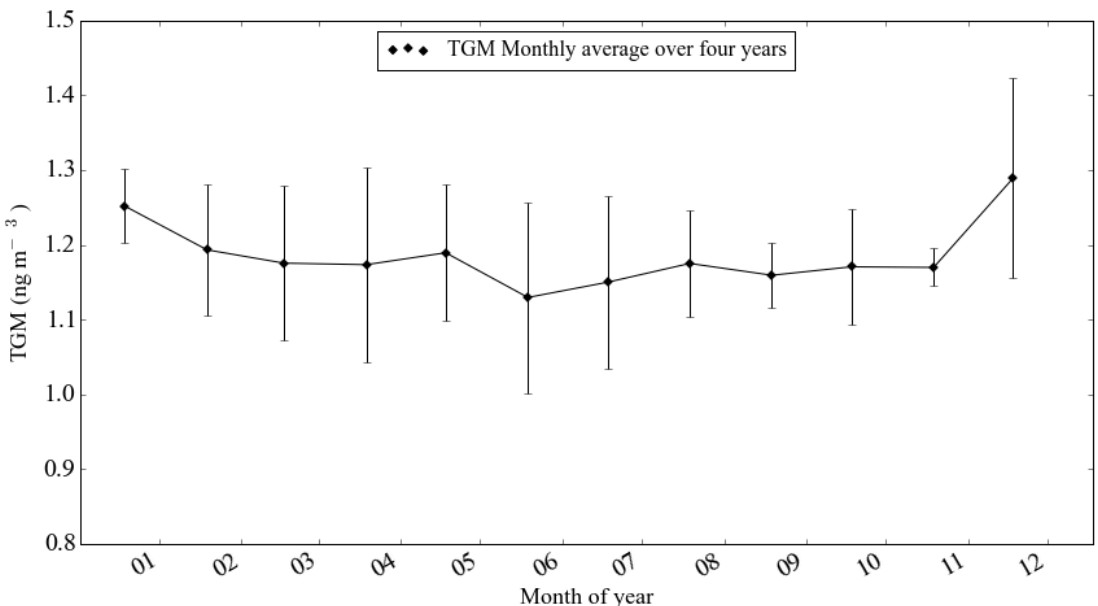

**Figure 3: Seasonal cycle of TGM at CVO. The bars represent the standard deviation of the monthly averages.**

Anthropogenic emissions of mercury affecting the Atlantic region include emissions from coal combustion, which tend to have maximum impact in February-March due to a dominance of air from continental regions such as North America. This is also observed in the seasonal distribution of anthropogenic combustion tracers such as carbon monoxide (Selin et al., 2007; Weigelt et al., 2015; Read et al., 2009). Ocean emissions of $Hg^0$ from the reduction of $Hg_{aq}^{II}$ to $Hg_{aq}^0$, driven by increased biological production are at a maximum in June in the NH but December in the SH (Strode et al., 2007). The seasonal trend may also be affected by meteorological differences in seasonal circulation patterns and cycles in boundary layer heights, clouds, precipitation and dry deposition characteristics (Dastoor and Larocque, 2004; Selin et al., 2007).

### 3.2 Four-year trends

The Theil-Sen function (Theil 1950; Sen 1968) was used to evaluate the 4-year dataset inclination based on monthly TGM medians by season. The results are shown in Fig. 4. In this function the slopes between all x, y pairs are calculated and the Theil-Sen estimate is the median of all these slopes. This analysis was performed using the Openair package in R (Carslaw et al., 2012). The advantage of using this function is that it gives accurate confidence intervals and is resistant to outliers. Statistics used for this plot can be found in Table 1 in the Supplementary information.

Over four years the data shows a weak downward inclination (0.042 ± 0.04 ng m$^{-3}$ yr$^{-1}$, p < 0.01 significance level). This decrease is more significant in the data collected during the Cape Verdean summer (-0.079 ± 0.054 ng m$^{-3}$ yr$^{-1}$, p <0.001 in June-August) than in the winter (-0.009 ± 0.179 ng m$^{-3}$ yr$^{-1}$, p < 0.1 in December-February). Previous studies have shown a stronger decreasing trend in air that has been influenced by anthropogenic emissions, for which there is more ready detection of the impact of regulation (Selin et al., 2007; Cole et al., 2014; Weigelt et al., 2015; Zhang et al., 2016).

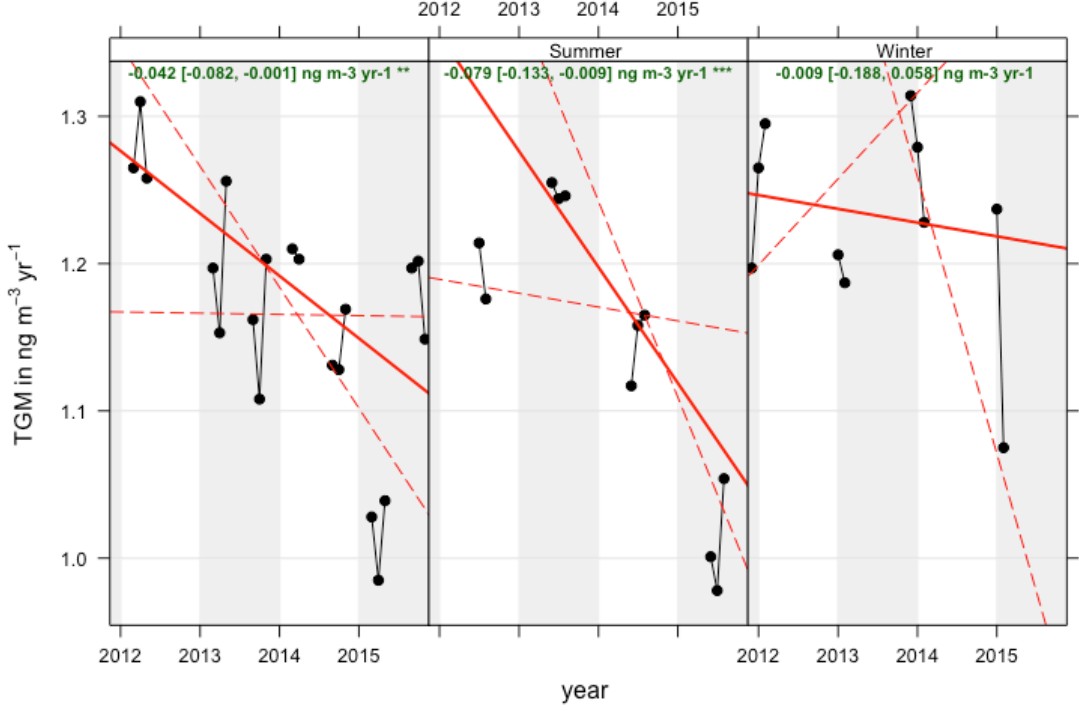

**Figure 4: TGM trends for the full year and then separated by season at the Cape Verde Observatory. The green text shows the slope estimate, with 99% confidence intervals in brackets.**

The seasonal trends calculated here imply that there are differences in the sources of mercury that affect the winter months compared to the summer months, potentially with a smaller decline in emissions over this winter period. This may be because CVO measures Hg coming from the same source region throughout the year but that the emission from that source has not declined as much in winter as it has in summer, e.g. from residential burning. Alternatively the difference could be explained by a difference in airmass between seasons bringing air from different sources that have experienced different trends in emissions over the years. We consider this latter scenario to be the more likely explanation, since air masses originating from continental Africa, which may be influenced by ASGM or biomass burning, frequently reach the CVO in winter (Carpenter et al., 2010), but are more rare in summer. A further alternative explanation for the difference in trends between seasons would be a change in global oxidant concentrations (such as OH) and that this effect had a seasonal dependence, but there is no evidence to support this from studies that estimate OH fields (Hartmann et al., 2013).

In order to understand better the drivers of the TGM behaviour, observations were classified according to the origin and pathways of air masses arriving at the CVO over a ten-day period using the UK Met Office NAME dispersion model in passive tracer mode (Ryall et al., 2001). The air mass classifications have been used previously for evaluating the source regions of reactive trace gases arriving at CVO (Carpenter et al., 2010). For this study eight geographical regions were defined (Coastal African, polluted Marine, Saharan Africa, Sahel Africa, North America, Atlantic marine South America, and, Tropical Africa) and from these, 7 air mass types are classified based on the percentage time spent over each of the 8 regions (Figures 5a and b). These are: Atlantic and African Coastal (AAC), Atlantic marine (AM), North American and Atlantic (NAA), North American and coastal African (NCA), European (with minimal African influence) (EUR), African (with minimal European influence) (AFR) and European and African (EUR/AFR). The eight regions are shown in Figure 5a and a trajectory frequency footprint of the trajectories (using all of the data from the measurement period), for each of the 7 classifications is shown in Figure 5b.

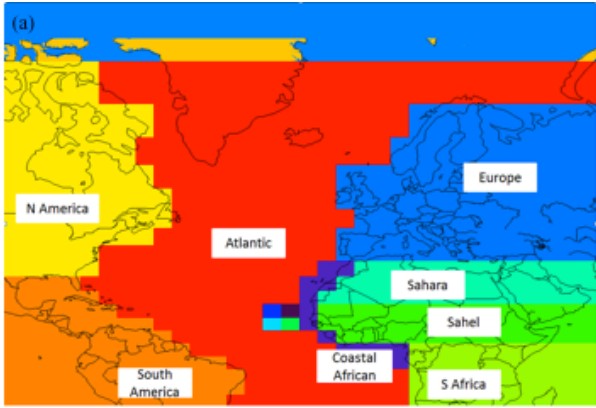

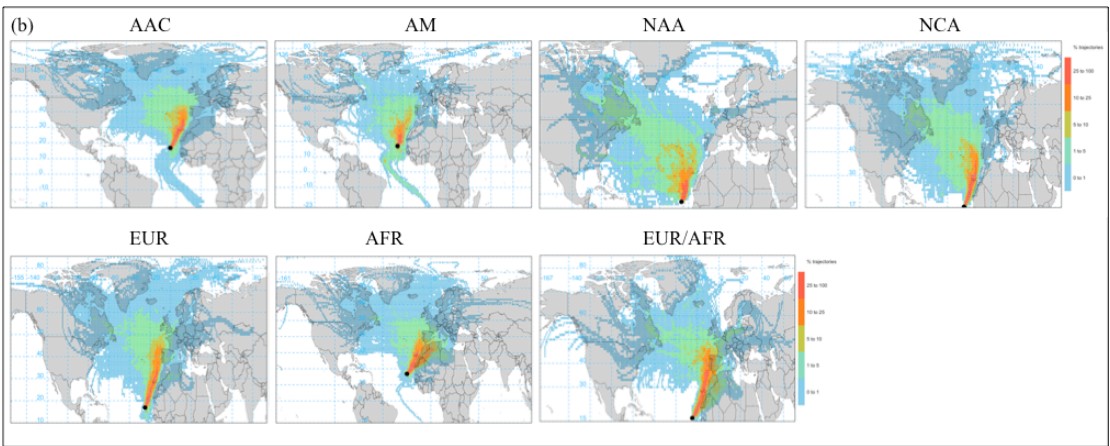

**Figure 5: a) Boundary definition of the eight geographical regions, Coastal African, polluted Marine,**
**Saharan Africa, Sahel Africa, North America, Atlantic marine South America, and, Tropical Africa. b)**
**Trajectory frequency maps for each of the seven air mass types using HYSPLIT trajectories and Openair,**
**AAC - Atlantic and African Coastal, AM - Atlantic Marine, NAA - North American and Atlantic, NCA -**
**North American and Coastal African, EUR - European (with minimal African influence), AFR - African**
**(with minimal European influence) and EUR/AFR - European and African.**
Figure 6 shows histograms representing the data in each of the 7 classifications and Table 2 details the
associated statistics.  The lowest variability in TGM was observed in air that had travelled the longest
period since contact with continental sources even though these would have been subjected to greatest
potential for ocean emissions (AM, NCA, NAA).  The lowest concentrations (1.144 +/- 0.109 ng m$^{-3}$)
were observed in Atlantic and African coastal air (AAC).  During late summer we occasionally (~2%
of time) receive air that has been influenced by the Southern hemisphere (Figure 5b and Supplementary
Information).  At these times the ozone mixing ratios drop to ~15 ppbV and there can be a rare
occurrence of rain leading to spikes in the mostly low TGM concentrations (Figure 7).  Otherwise, the
highest and most variable concentrations of total gaseous mercury (mean of 1.23 ± 0.16 ng m$^{-3}$) are
observed in air originating from continental Africa (AFR).

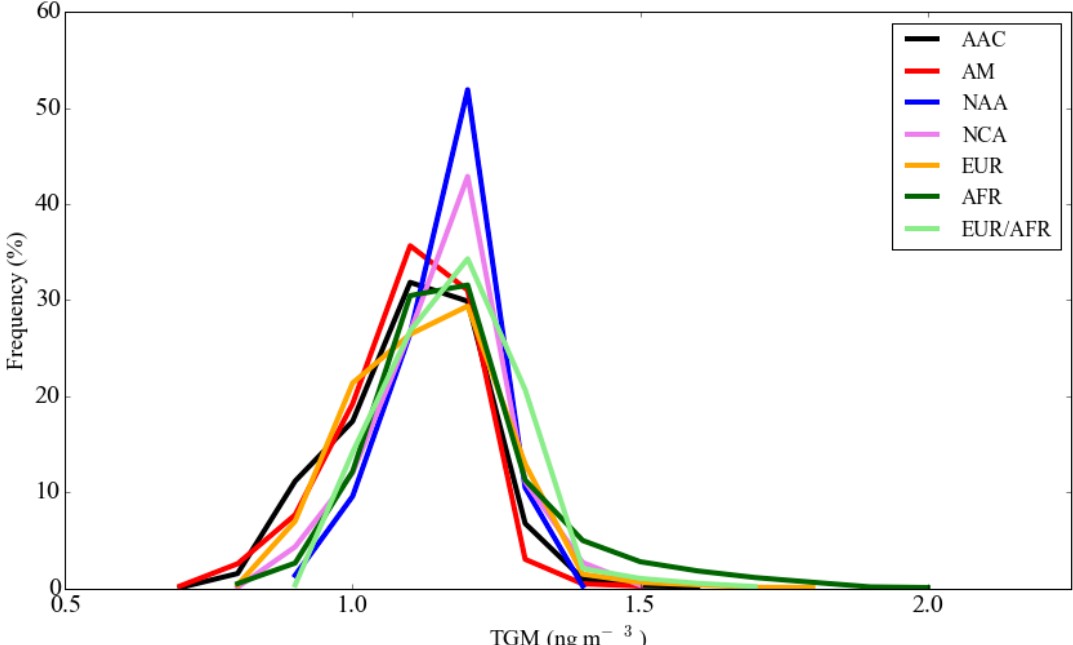

**Figure 6: Histograms of observed TGM classified by airmass for the full 2011-2015 dataset.**

| Air mass | Mean +/- 1 sigma standard deviation (25th-75th percentiles, number of points) ng m-3 | % Time the site receives air mass |
|---|---|---|
| AM | 1.14 +/- 0.11 (1.08-1.22, 432) | 6% |
| AAC | 1.15 +/- 0.12 (1.08-1.24, 1633) | 24% |
| NAA | 1.21 +/- 0.08 (1.17-1.27, 449) | 7% |
| NCA | 1.20 +/- 0.10 (1.15-1.27, 975) | 15% |
| EUR | 1.18 +/- 0.13 (1.08-1.26, 1161) | 17% |
| AFR | 1.23 +/- 0.16 (1.14-1.29, 1448) | 22% |
| EUR/AFR | 1.22 +/- 0.11 (1.16 -1.30, 586) | 9% |

**Table 2. Statistics for the individual air mass classified data**.
Paired t-tests were performed (Wilcoxon signed rank test, R) using the air mass datasets and the AFR
dataset showed a significant difference (significance level <95%) in the mean concentration when
compared to AAC, NAA, NCA, EUR, EUR/AFR and AM.  This suggests that the air classified as AFR
may be influenced by sources with different longer-term emissions trends to those experienced when
other air masses are detected (t-test results can be found in Table 2 in the Supplementary Information).
Biomass burning, of both anthropogenic and biogenic origins, is prevalent in Africa.  In the Northern
Hemisphere Africa burning occurs primarily in the Sahel, moving from the northern to the southern
Sahel between November and February (Roberts et al., 2009).  From Figure 5b it would appear that
there are few trajectories which originate from this region, however an influence from biomass burning
could be one explanation for the variable and sometimes higher, mercury concentrations within AFR
air masses.  Previous studies have found a relationship between TGM and carbon monoxide during
such episodes (Slemr et al., 2006, Brunke et al, 2012). Figure 7 shows a correlation analysis using a
matrix method between pairs of data at the CVO for AFR air, separated by season.  $O_3$ and CO show a
strong positive correlation particularly in spring, but also in summer and winter consistent with their
shared pollution sources and of CO being a precursor of $O_3$ over long transport times.  Higher wind
speeds tend to be associated with air masses that have travelled further (from continental regions), and
therefore have undergone greater photochemical production, which might explain the positive
correlation of $O_3$ with wind speed.  In spring a strong positive correlation of TGM with CO and with
$O_3$ suggests a shared anthropogenic source when these concentrations are at their seasonal high in the
northern hemisphere.  The lack of correlation of $O_3$ with CO in autumn however suggests a more
localised source for CO, perhaps from biomass burning, but this is not reflected in a positive

relationship with the TGM concentrations. Instead, TGM has a strong correlation with NOx (NO + NO$_2$), which suggests that the TGM concentrations are influenced by anthropogenic pollution sources within closer proximity, for example from pollution emitted from cities on the coast of West Africa. This may also explain the high variability in the TGM concentrations observed in the air classified as AAC, which has also travelled over the African coast; although the concentrations in that air mass are lower due to longer time spent over the ocean. Air classified as AFR and EUR/AFR shows the highest levels of TGM and these may be better explained by an additional source further in-land. If sources of mercury from small-scale artisanal gold mining (ASGM) from West Africa have any impact on these measurements and trends then this may be reflected in a weaker correlation with CO in autumn and winter. This is because we speculate that the activity might be more commonly carried out in the dry season (November to April) when crops can't be grown; however there is little evidence to support this. There is no data for TGM in AFR in summer.

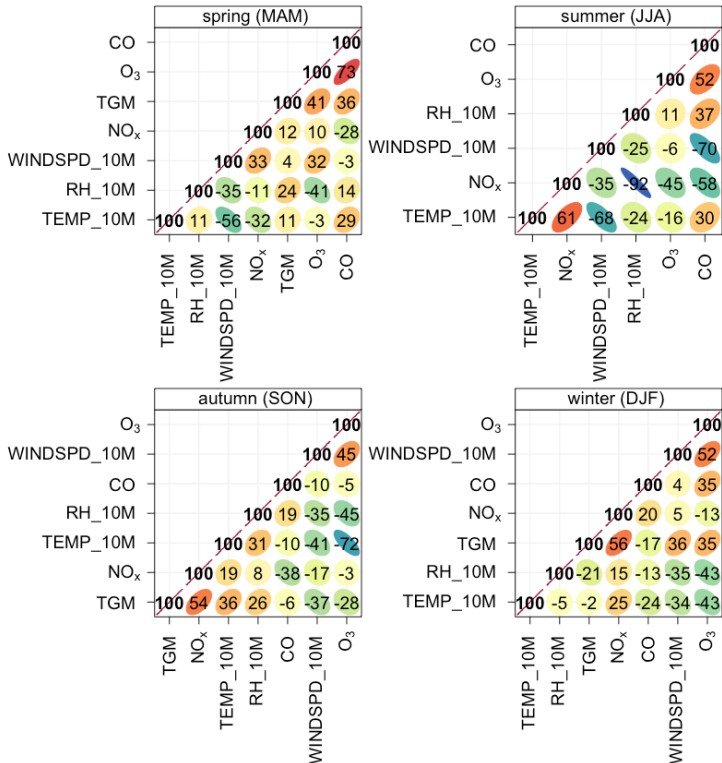

**Figure 7: A correlation matrix separated by season to show the correlation between pairs of data, using the corPlot function in Openair (Carslaw et al., 2012). The ellipses are visual representations of a scatter plot. The colour scale highlights the strength of the correlation (red being the strongest and blue the weakest), and the number is the r$^2$ of the data. The data was daily averaged before correlating to remove any bias from diurnal variability. The order the variables appear is due to their similarity with one another, through hierarchical cluster analysis**

We next consider episodes when TGM concentrations were enhanced, to investigate the potential influence of biomass burning on the measurements. On two occasions TGM exceeded 1.7 ng m$^{-3}$ (~0.5 ng m$^{-3}$ higher than the mean levels detailed in Table 1). The first period was during December 2013; Figure 8a shows normalised CO and TGM concentrations during this month. The shaded periods correspond to AFR trajectories, shown in detail in Figure 8b. The figure shows that for this period of relatively elevated concentrations in AFR classified air, there is no significant correlation with CO, nor do the trajectories originate over the biomass-burning region of the Sahel. Thus, biomass burning appears not to be a contributor to the high TGM concentrations during this period.

(a)

(b)

Figure 8: (a) Time-series of TGM and carbon monoxide. The plot is normalised by dividing by the compounds' mean value and the grey dots indicate when the air was characterized as AFR. The shaded periods correspond to the 10-day back trajectories in (b). Clockwise from top left: 15[th] December 2013 03:00, 17[th] December 2013 18:00, 23rd December 2013 03:00, and 20[th] December 2013 00:00.

A similar analysis was performed for the period 19[th] September 2015 until the 19[th] October 2015 and the corresponding plots and trajectories are shown in Figure 9. In this case the period of elevated concentrations is shorter with the episode lasting around a week. From the trajectories the air may have been influenced by air from the biomass region in Sahel Africa, which is at its most northern location in the month of October (Roberts et al., 2009). However, CO was not elevated during the period of peak [TGM] suggesting that biomass burning was not the source.

(a)

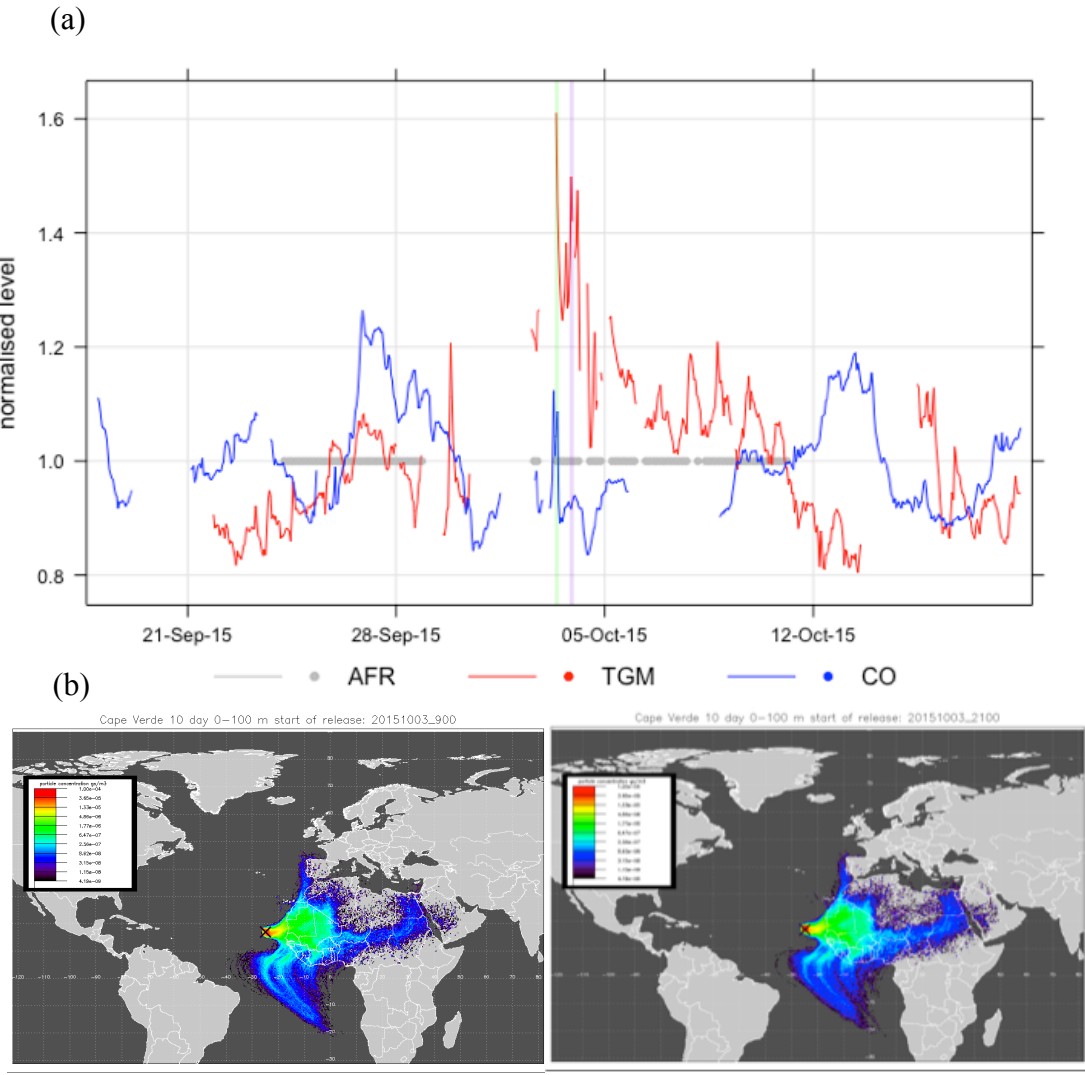

(b)

**Figure 9: a) Time-series of TGM and CO. The plot is normalised by dividing by the compounds mean value and the grey dots indicate when the air was characterized as AFR.  The shaded periods correspond to the 10-day back trajectories in b), left, 3$^{rd}$ October 2015 9:00 and right, 3$^{rd}$ October 2015 21:00.**

It has been previously established that West Africa is an important source region for ASGM activity (Telmer and Velga, 2009) but it is difficult to determine whether West Africa is a growing source of emissions since data has been limited and is subject to large uncertainty.  It is likely however that the ASGM emissions are less regulated than anthropogenic emissions from coal combustion, ferrous/non-ferrous metal and cement production from Europe and the US (UNEP, 2013), and so is less likely to be decreasing in source strength.

A further analysis using the Theil-Sen function was performed to evaluate trends in the 4-year dataset within individual air masses, based on seasonal TGM medians (Figure 10).

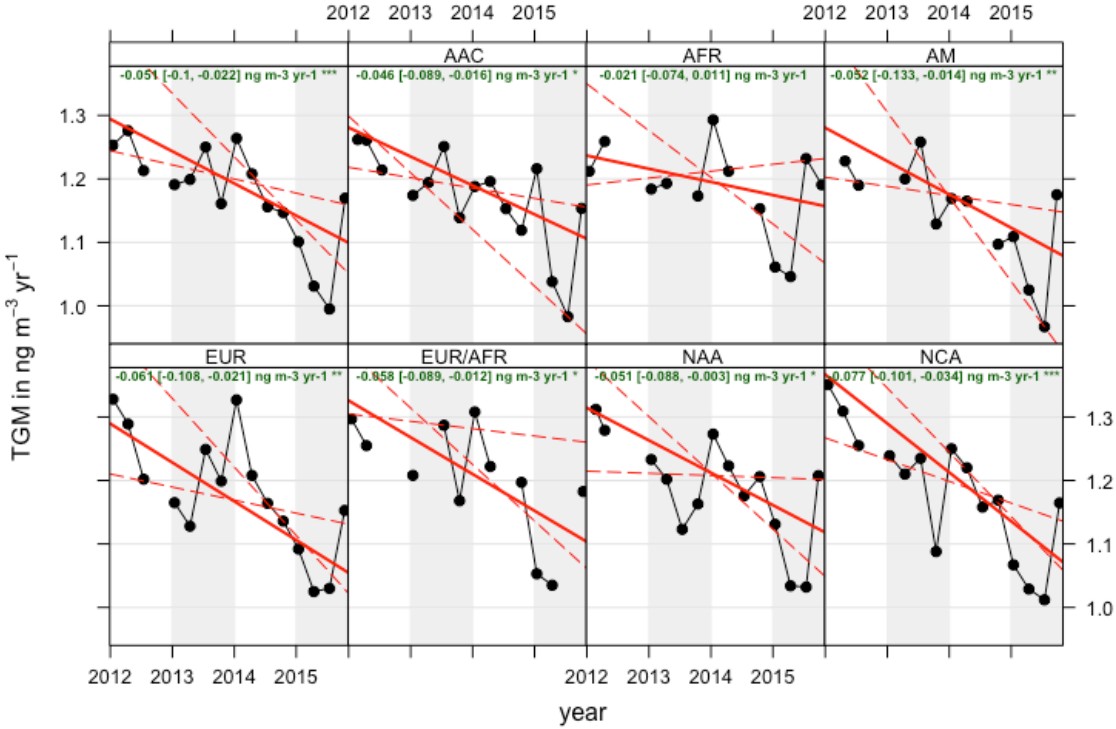

**Figure 10: 4-year TGM trends using a Theil-Sen function based on seasonal TGM medians separated by air mass. The data for all air masses is shown in the top left panel with the rest of the panels showing the data separated by the seven air mass classifications. The green text shows the slope estimate and the 95% confidence intervals are in brackets. In each case the solid red line shows the trend estimate and the dashed red lines show the 95% confidence intervals for the trend based on resampling methods.**

The overall data (top left panel) show a decrease of -0.051 ng m$^{-3}$ yr$^{-1}$ (95% confidence interval of -0.1
to -0.022 ng m$^{-3}$ yr$^{-1}$), as shown in the green text. Over the 4-year period the concentration changes
were (in ng m$^{-3}$ yr$^{-1}$): -0.046 ± 0.037, -0.021 ± 0.043, -0.052 ± 0.060, -0.061 ± 0.044, -0.058 ± 0.039, -
0.051 ± 0.043, -0.077 ± 0.034, for AAC, AFR, AM, EUR, EUR/AFR, NAA and NCA respectively
(See Table 3 in the Supplementary information for the number of points used to derive monthly
medians). The symbol "***" in green indicates that the decrease is significant to the 0.001 level, "**"
to 0.01 and "*" 0.05. Decreasing trends were observed in all the air masses with the largest trends
observed in NCA and EUR suggesting controls on anthropogenic emissions are having an effect.
Although there is some overlap in the 95% confidence intervals, the African (AFR) air clearly shows
the smallest decreasing trend and the only positive upper confidence interval over the 4 years compared
to all of the other air masses.
From an analysis of trajectories it was found that the AFR air was within the Sahel region outlined
earlier for only around 1-2% of the time and mostly in October when the burning is at its most northern
point (Figure 5b). In order to evaluate whether these air masses biased the decrease in AFR air, a
sensitivity analysis was performed on the ThielSen analysis with the October data removed. The data
flagged as southerly was also removed. The updated concentration change was -0.022 ± 0.036 ng m$^{-3}$
19  yr$^{-1}$, thus the filtering had essentially no effect on the AFR trend.
Weigelt et al., (2015) observed an annual decrease in TGM concentrations at Mace Head (53°20'N,
9°54'W, 10 m asl) of between -0.021 and -0.023 ng m$^{-3}$ yr$^{-1}$ between 1996-2013, close to the upper
confidence levels shown here (Weigelt et al., 2015). Within sub-tropical maritime classified air masses
(air from south of 28°N, west of 10°W) the decrease was -0.016 ± 0.002 ng m$^{-3}$ yr$^{-1}$. Further south at
Cape Point, the trend for 17 years (1996-2013) was reported as -0.018 ng m$^{-3}$ yr$^{-1}$ (95% significance
interval of -0.035 to -0.013 ng m$^{-3}$ yr$^{-1}$). These concentration decreases are lower than the global
calculation using the GEOS-CHEM model for the Northern Atlantic of -0.04 ng m$^{-3}$ yr$^{-1}$ (Soerensen et
al., 2012), but may not take into account the very recent decreases in emissions (Weigelt et al., 2015;
Soerensen et al., 2012). The overall concentration decrease (2011-2015) observed here of -0.05 ng m$^{-3}$
30  yr$^{-1}$ is more similar to the GEOS-CHEM study.

## 4 Conclusions

We report a four-year decreasing trend in total gaseous mercury (TGM) concentrations over the sub-tropical north Atlantic of -0.05 ± 0.04 ng m$^{-3}$ yr$^{-1}$, a rate of decline that is in agreement with a GEOS-CHEM analysis for the Northern Atlantic (Soerensen et al., 2012). A downward trend in concentration was observed in 6 out of 7 different air mass types, all associated broadly with long-range transport of air from the US and Europe over the north Atlantic ocean to the measurement location. The smallest and least significant downward trend (-0.02 ± 0.03 ng m$^{-3}$ yr$^{-1}$) was observed in air that was influenced by West Africa, where emissions are less understood and may well be static or possibly still increasing. The UNEP Global Mercury Assessment report in 2013 suggested that more work is needed to improve emissions estimates for West African sources including field measurements around artisanal and small-scale gold mining (ASGM) sites.

## Data availability

The data presented here is freely available at the Centre for Environmental Data Analysis (CEDA) at http://catalogue.ceda.ac.uk/uuid/0ae5eb7ce3ad4885a7223dd7b69f4db6. Other levels of data are available within the GMOS central database upon request at http://sdi.iia.cnr.it/geoint/publicpage/GMOS/gmos_historical.zul (GMOS Database, 2014).

## Author contribution

K. A. Read, L.J Carpenter, A. C. Lewis, J. Kentisbeer contributed to the preparation of the manuscript.
K. A. Read and L.M. Neves made the measurements
Z. Fleming ran the NAME trajectories and assigned the air mass classifications.

## Acknowledgements

The authors acknowledge the Natural Environmental Research Council (NERC) and the Atmospheric Measurement Facility (AMF), National Centre for Atmospheric Science (NCAS) for their continued funding of the Cape Verde Observatory. The measurements of TGM were initiated due to financial support from the EU FP7-ENV-2010 project "Global Mercury observation System" (GMOS, Grant Agreement no 265113).

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
