# Peer review of "Four years (2011-2015) of Total Gaseous Mercury Measurements from the Cape Verde Atmospheric Observatory"

_Atmospheric Chemistry and Physics, 2016_

## Referee Comment (RC1) · Anonymous Referee #1 · 19 Dec 2016

This paper is very well written, the information is presented in a clear and precise manner and the authors present and back up their arguments in an exhaustive manner.

I recommend this paper for publication as is.

---

## Referee Comment (RC2) · Anonymous Referee #2 · 21 Dec 2016

The authors present measurements of mercury concentrations in 2011 - 2015 at a coastal site located at Cabo Verde archipelago. They used a standard, well established, quality controlled technique. The data are analysed mainly in terms of comparison with measurements at other sites around the Atlantic Ocean, seasonal variation, trends and their seasonal variation, differences between air masses of different origins, and diurnal variations.

The data are valuable and I recommend a publication of their analysis. Unfortunately, the presented analysis is rather superficial and at times flawed. The used statistical methods are not always described and the statistical significance of averages, trends and their differences is frequently not given. Consequently, probably insignificant difcreate

ferences are sometimes discussed at length. In addition, the discussion is at times muddled by using false references or misusing some. The major deficiency of the paper is, however, that a suite of other species is measured at Cabo Verde, such as O3, CO, nitrogen oxides, VOCs, and greenhouse gases, even BrO, but these data are completely ignored in the presented analysis. E.g. CO data could provide decisive information about the origin of high Hg concentrations from the African continent: whether it is biomass burning or small scale artisanal gold mining. The BrO data could provide important clues about bromine chemistry. The ancillary data could also help to identify air masses from the southern hemisphere. In addition, they could help (with backward trajectories) to explain the origin of events with extremely high and extremely low mercury concentrations. As it is, the mercury data are used far below their potential. I recommend the publication of the paper after substantial improvements, some of which are listed below.

Specific comments

Page 1, line 38: GEM has to be defined when introduced. GEM is Gaseous Elemental Mercury and as such not TGM. TGM is GEM + RGM.

Page 1, line 41-42: In the context of this paper, biomass burning is probably a very important Hg source. It may be initiated by man or nature and as such does not fit only the category of anthropogenic sources.

Page 2, line 38: The paper by Slemr et al (2013) as cited in references is about 222Rn calibrated terrestrial fluxes in southern Africa and not about trends. A paper by Slemr et al about trends would be Atmos. Chem. Phys. 11, 4779-4787, 2011.

Page 3, line 15: "Leipzig" instead of "Leibzig"

Page 3, line 30, to page 4, line 8: It should be mentioned that GEM is measured because RGM (or GOM) will be most likely captured by the salt deposited at the inlet tubing and the particle filter. The standard conditions of the reported mercury concentrations have also to be clearly stated.

Page 3, line 18-26: Information about diurnal circulation pattern (sea and land breeze) is needed because it is important for the interpretation of the diurnal cycle.

Page 5, line 16-18: Chemistry is not the only possible explanation, an influence of air masses from the southern hemisphere without any pronounced seasonal variation may be another (Slemr et al., Atmos. Chem. Phys. 15, 3125-3133, 2015).

Figure 3: What do the bars represent?

Page 5, line 24-26: Duncan et al. (2007) do not say anything about seasonal variation of specifically emissions from coal burning - they calculate only emissions from fossil fuels. In fact, emissions from coal burning tend to have a flat seasonal variation because of residential heating in winter and air conditioning in summer (Rotty, Tellus 39B, 184-202, 1987). The sentence needs rewording.

Section 3.2 about trends: The method of trend calculation is not clearly stated and neither is the significance of the discussed trends. Non-significant trends should not be discussed at all. The significance of the presented trends is probably low because only 4 years of measurements with numerous gaps are available. In addition, the unevenly distributed data gaps and pronounced longer term events with high GEM concentration, such as after December 2013 and before December 2015 could produce wrong trends. A defensible statistical method of calculation of trends and their significance is sorely needed. When stating averages and medians, the number of measurements they represent has always to be stated. Without the number of measurements no statistical tests for significance of differences can be made.

Page 6, line 8: The trend has to be given with 3 significant decimal numbers to be consistent with its standard deviation.

Page 7: For the regional classification of origin of air masses the reader has now to consult Carpenter et al. (2010). A reproduction of the figure from 5 from Carpenter et

al. (2010) or even better a similar figure of trajectories for 2011 – 2015 would make the reading more comfortable.

Figure 5: The colour differences are not very pronounced, please improve.

Table 2: Number of measurements should be stated for each average.

Page 7, line 12-23: This discussion would make sense only if the average concentration for AFR air masses were significantly different from the other air masses. Statistical test for differences have thus to be presented. As mentioned before, biomass burning has to be considered in addition to the AGSM activity and their relative importance could be assessed by using the CO data.

Page 8, line 9-12: The statement that the method is "resistant to outliers" is not quite true because monthly averages themselves can be strongly influenced by outliers. Using monthly medians instead of monthly averages would remove this objection. It could even provide more certain results – see below.

Figure 6: The slopes and their uncertainty in green have to be substantially enlarged in final version to become readable.

Page 8, line 16-24: I think that this discussion is flawed. All the calculated trends including the one for AFR are significant at some numerical level, the significance for AFR being the lowest but still existent. In other words: the AFR trend is more uncertain than the other ones but still existing. What matters for the discussion are the differences between the AFR trend and other trends (their slopes), not the absence of the significant AFR trend at a preselected discreet significance level of 95% (why not 90% or any other number?). Because of the larger uncertainty of the AFR trend there is probably no significant difference between the trend slopes. If so, there is no difference to be discussed. The discussion also neglects that the AFR trend is based on the smallest number of monthly averages (the number of monthly averages for each trend should be given in Figure 6) one of which is with 1.6 ng m-3 extremely high.

[Figure]

Page 8, line 26-34: The discussion in this paragraph is muddled. In the first sentence measurement at Mace Head (Weigelt et al., 2015) are compared with those at Cabo Verde which is questionable because of geographical and temporal difference. The trend of -0.016 $\pm$ 0.002 ng m-3 yr-1 for subtropical maritime air masses is implausibly precise for a suggestion originating from a model by Soerensen et al. (2012). In fact this trend originates from the analysis of Weigelt et al (2015) specifically for subtropical maritime air masses, not from Soerensen et al. According to Weigelt et al. (2015) this trend did not changed with time and can thus be directly compared with measurements at Cabo Verde in 2011-2015. No levelling off was observed for these air masses in contrast to the statement in the last sentence of this paragraph. In the third sentence the trend is erroneously compared with seasonal variation.

Section 3.3 Short term variability: With statistically insignificant diurnal variation it does not make much sense to discuss the difference between a maximum and a minimum. To reveal a more distinct diurnal variation one should try to get rid of the day-to-day variation by e.g. normalizing the hourly data to a daily average. Because Br is also produced photochemically the diurnal variation should not be discussed solely in terms of OH chemistry. I wonder why BrO measurements at the station are not used in the interpretation of the mercury chemistry. As a coastal site, the diurnal variation may also be influenced by sea and land breeze.

---

## Author Comment (AC1) · 15 Mar 2017

Response to both reviewers

The authors thank the reviewers for both their time and comments. We have made some major changes to the manuscript to address comments regarding the significance of the trend analysis, the inclusion of other datasets from the CVO, the influence of biomass on the TGM measurements and the impact of southern hemispheric air on the site.

General comments

COMMENT:The data are valuable and I recommend a publication of their analysis.

Unfortunately, the presented analysis is rather superficial and at times flawed. The used statistical methods are not always described and the statistical significance of averages, trends and their differences is frequently not given. Consequently, probably insignificant diferences are sometimes discussed at length. In addition, the discussion is at times muddled by using false references or misusing some. The major deficiency of the paper is, however, that a suite of other species is measured at Cabo Verde, such as O3, CO, nitrogen oxides, VOCs, and greenhouse gases, even BrO, but these data are completely ignored in the presented analysis. E.g. CO data could provide decisive information about the origin of high Hg concentrations from the African continent: whether it is biomass burning or small scale artisanal gold mining. The BrO data could provide important clues about bromine chemistry. The ancillary data could also help to identify air masses from the southern hemisphere. In addition, they could help (with backward trajectories) to explain the origin of events with extremely high and extremely low mercury concentrations. As it is, the mercury data are used far below their potential. I recommend the publication of the paper after substantial improvements, some of which are listed below.

RESPONSE: Specifically we have focused on the potential influences on the TGM measurements in the AFR classified air, since this is the region subject to most variability and smallest decreasing trend; and we have considered the correlations of a number of species (O3, CO, NOx, meteorological factors) with TGM and with each other. We have found that there is a lack of correlation in autumn and winter of CO with TGM which suggests different sources for these species at a time when both biomass burning and emissions from ASGM are likely at their seasonal highest. We note that for the most part TGM correlates with NOx inline with an anthropogenic source (coal burning, cement production, oil refining etc) in West Africa. We have included a figure (Figure 7) to illustrate the correlations along with some additional discussion (p.9, l.13-p.10, lns1-13, revised manuscript). We have focused on two time periods when there are elevated TGM concentrations (>1.7 ng m-3)) and used back trajectory analysis and ancillary data to determine the potential origins of the TGM (p.10, l.23-p12, revised

[Figure]

manuscript). We do not see evidence for biomass burning at either of these times and suggest that the increased variability and smaller decreasing trend on top of the overall background concentrations (from anthropogenic city emissions) is due to an additional unregulated source from ASGM in this region. We have not used any BrO measurements in our revised analysis since we have no measurements available during the period of TGM measurements discussed within this paper. Additional measurement detail has been included in the experimental section regarding other datasets (p.4, lns 19-32, revised manuscript). More statistical information regarding the plots has been included into a supplementary information section. Finally seven new references have been included, and one deleted.

Specific comments

COMMENT:Page 1, line 38: GEM has to be defined when introduced. GEM is Gaseous Elemental Mercury and as such not TGM. TGM is GEM + RGM.

RESPONSE: Page 1, line 38: GEM has been defined.

COMMENT:Page 1, line 41-42: In the context of this paper, biomass burning is probably a very important Hg source. It may be initiated by man or nature and as such does not fit only the category of anthropogenic sources.

RESPONSE:Page 1, line 41-42: A short description of the importance of biomass burning has been added.

COMMENT:Page 2, line 38: The paper by Slemr et al (2013) as cited in references is about 222Rn calibrated terrestrial fluxes in southern Africa and not about trends. A paper by Slemr et al about trends would be Atmos. Chem. Phys. 11, 4779-4787, 2011.

RESPONSE:Page 2, line 38: This reference has been corrected.

COMMENT:Page 3, line 15: "Leipzig" instead of "Leibzig"

RESPONSE:Page 3, line 15: Leipzig has been spelt correctly.

COMMENT:Page 3, line 30, to page 4, line 8: It should be mentioned that GEM is measured because RGM (or GOM) will be most likely captured by the salt deposited at the inlet tubing and the particle filter. The standard conditions of the reported mercury concenC2 ACPD Interactive comment Printer-friendly version Discussion paper trations have also to be clearly stated.

RESPONSE:Page 3, line 30: These technical points have been stated.

COMMENT:Page 3, line 18-26: Information about diurnal circulation pattern (sea and land breeze) is needed because it is important for the interpretation of the diurnal cycle. RESPONSE:Page 3, line 18-26: Some comments about the diurnal circulation pattern at the site have been included into the Experimental section (page 3 new lines 29-32). The section on Short term variability (Section 3.3) has been removed.

COMMENT:Page 5, line 16-18: Chemistry is not the only possible explanation, an influence of air masses from the southern hemisphere without any pronounced seasonal variation may be another (Slemr et al., Atmos. Chem. Phys. 15, 3125-3133, 2015).

RESPONSE:Page 5, line 16:18 Air mass back trajectory analyses shows that the CVO receives very little air representative of the southern hemisphere (∼1.3% of all data, Fig. 1. Supplementary Information). Further, the highest frequency of southerly air masses arriving at the CVO occurs during August and September, which would serve to increase the mercury seasonal cycle amplitude rather than reduce it. Discussion has been added to the text (p.6, lns 1-6, revised manuscript).

COMMENT:Figure 3: What do the bars represent?

RESPONSE:Figure 3: An explanation for the bars has been added and the plot scale improved.

COMMENT:Page 5, line 24-26: Duncan et al. (2007) do not say anything about seasonal variation of specifically emissions from coal burning - they calculate only emissions from fossil fuels. In fact, emissions from coal burning tend to have a flat seasonal

variation because of residential heating in winter and air conditioning in summer (Rotty, Tellus 39B, 184-202, 1987). The sentence needs rewording.

RESPONSE:Page 5: line 24-46: This sentence has been reworded (p.6, lns 10-13, revised manuscript).

COMMENT:Section 3.2 about trends: The method of trend calculation is not clearly stated and neither is the significance of the discussed trends. Non-significant trends should not be discussed at all. The significance of the presented trends is probably low because only 4 years of measurements with numerous gaps are available. In addition, the unevenly distributed data gaps and pronounced longer term events with high GEM concentration, such as after December 2013 and before December 2015 could produce wrong trends. A defensible statistical method of calculation of trends and their significance is sorely needed. When stating averages and medians, the number of measurements they represent has always to be stated. Without the number of measurements no statistical tests for significance of differences can be made.

RESPONSE:Section 3.2: The box and whisker plots have been replaced with a Thiel-sen analysis for consistency with the later air mass analyses, and reference has been made to the significance of the trends. The number of measurements used to calculate the medians has been included into the Supplementary information, Table 1.

COMMENT:Page 6, line 8: The trend has to be given with 3 significant decimal numbers to be consistent with its standard deviation.

RESPONSE:Page 6, line 8: The Thiel-sen calculated values are now quoted to 3 decimal places.

COMMENT:Page 7: For the regional classification of origin of air masses the reader has now to consult Carpenter et al. (2010). A reproduction of the figure from 5 from Carpenter et C3 ACPD Interactive comment Printer-friendly version Discussion paper al. (2010) or even better a similar figure of trajectories for 2011 – 2015 would make the

reading more comfortable.

RESPONSE:Page 7: We agree. We have added a new Figure 5a which shows the eight geographical regions used for the analysis and b) 10-day trajectory footprints for the measurement period as frequency maps for each of the air masses.

COMMENT:Figure 5: The colour differences are not very pronounced, please improve.

RESPONSE:Figure 5: The colours on this plot have been improved (Figure 6 in revised manuscript).

COMMENT:Table 2: Number of measurements should be stated for each average.

RESPONSE::Table 2: The number of measurements for each median has now been included in Table 2. Some errors in this table have also been remedied.

COMMENT:Page 7, line 12-23: This discussion would make sense only if the average concentration for AFR air masses were significantly different from the other air masses. Statistical test for differences have thus to be presented. As mentioned before, biomass burning has to be considered in addition to the AGSM activity and their relative importance could be assessed by using the CO data.

RESPONSE:Page 7, lines 12-23: Paired t-tests have been performed on the air mass datasets and the AFR air was found to be statistically different from the other air masses (text added to pg 10 lns 4-8). Additional case study type analysis using trajectories and correlations with measurements of carbon monoxide and ozone have been performed with the inclusion of new Figures 7, 8, and 9. More discussion with respect to an alternative source from biomass burning has now been included into the text (p.10, l.23-p12, revised manuscript).

COMMENT:Page 8, line 9-12: The statement that the method is "resistant to outliers" is not quite true because monthly averages themselves can be strongly influenced by outliers. Using monthly medians instead of monthly averages would remove this objection. It could even provide more certain results – see below.

RESPONSE:Page 8, line 9-12: The analysis has been improved by using medians because of the potential effect of outliers (and missing data) on the mean values and the data averaged by season rather than by month to improve the statistics. This is the new Figure 10. The 4-year decrease is still lower in the AFR classified air and the shape of the plot is very similar, which suggests the influence is not episodic but widespread. The levels are generally higher in that air mass compared to other air masses in the later years. The number of points used to calculate the monthly medians has been included in Table 3 in the Supplementary information.

COMMENT:Figure 6: The slopes and their uncertainty in green have to be substantially enlarged in final version to become readable.

RESPONSE:Figure 6: The text has been enlarged.

COMMENT:Page 8, line 16-24: I think that this discussion is flawed. All the calculated trends including the one for AFR are significant at some numerical level, the significance for AFR being the lowest but still existent. In other words: the AFR trend is more uncertain than the other ones but still existing. What matters for the discussion are the differences between the AFR trend and other trends (their slopes), not the absence of the signifi- cant AFR trend at a preselected discreet significance level of 95% (why not 90% or any other number?). Because of the larger uncertainty of the AFR trend there is probably no significant difference between the trend slopes. If so, there is no difference to be discussed. The discussion also neglects that the AFR trend is based on the smallest number of monthly averages (the number of monthly averages for each trend should be given in Figure 6) one of which is with 1.6 ng m-3 extremely high.

RESPONSE:Page 8, line 16-24: The discussion has been rewritten to concentrate on the differences between air mass trends rather than on the individual significance of the air masses at a specific level (p.13, lns 3-13, revised manuscript). We have included a +/- value on each trend using the data from the 95% confidence intervals. In fact it is the AM trend, which shows the largest uncertainty, rather than the AFR

trend. Although there is some overlap in the confidence intervals, the AFR data shows the smallest trend and the highest (only positive) upper confidence interval. In the revised analysis we use seasonal medians rather than monthly averages, and the same number of points is used for the AM and AFR analysis whilst one less median is used for the EUR/AFR trend. These statistics have been included into the Supplementary information.

COMMENT:Page 8, line 26-34: The discussion in this paragraph is muddled. In the first sentence measurement at Mace Head (Weigelt et al., 2015) are compared with those at Cabo Verde which is questionable because of geographical and temporal difference. The trend of -0.016 $\pm$ 0.002 ng m-3 yr-1 for subtropical maritime air masses is implausibly precise for a suggestion originating from a model by Soerensen et al. (2012). In fact this trend originates from the analysis of Weigelt et al (2015) specifically for subtropical maritime air masses, not from Soerensen et al. According to Weigelt et al. (2015) this trend did not changed with time and can thus be directly compared with measurements at Cabo Verde in 2011-2015. No levelling off was observed for these air masses in contrast to the statement in the last sentence of this paragraph. In the third sentence the trend is erroneously compared with seasonal variation.

RESPONSE: Page 8, line 26-34: The text in this section has been rewritten to make it clearer (p.13, lns 22-31, revised manuscript), the Sorensen model was not well referenced although it was correct for the model result that was quoted.

COMMENT: Section 3.3 Short term variability: With statistically insignificant diurnal variation it does not make much sense to discuss the difference between a maximum and a minimum. To reveal a more distinct diurnal variation one should try to get rid of the day-to-day variation by e.g. normalizing the hourly data to a daily average. Because Br is also produced photochemically the diurnal variation should not be discussed solely in terms of OH chemistry. I wonder why BrO measurements at the station are not used in the interpretation of the mercury chemistry. As a coastal site, the diurnal variation may also be influenced by sea and land breeze.

[Figure]

RESPONSE: Section 3.3: This section has been removed, as it needs more explanation and analysis, which is beyond the scope of this paper.

Please also note the supplement to this comment:
http://www.atmos-chem-phys-discuss.net/acp-2016-1036/acp-2016-1036-AC1-supplement.pdf

**Supplement:**

**Supplementary information**

[Figure]

Figure 1: Cluster analysis of data collected between December 2011- December 2015 into 7 clusters using the trajCluster function in Openair (Carlaw et al., 2012).

[Figure]

Figure 2: Ozone measurements from the CVO plotted with those from Mace Head, Ireland and Tudor's Hill, Bermuda for the same period.

The seasonal cycle of ozone at CVO and at Mace Head are similar except in Summer when the site is more influenced by southern air and the concentrations at the CVO are lower.

| Month | 2011-2012 | 2012-2013 | 2013-2014 | 2014-2015 |
|---|---|---|---|---|
| December | 510 | 0 | 714 | 1 |
| January | 634 | 203 | 695 | 307 |
| February | 690 | 541 | 673 | 625 |
| March | 721 | 744 | 744 | 628 |
| April | 717 | 133 | 372 | 96 |
| May | 514 | 411 | 0 | 591 |
| June | 0 | 721 | 109 | 493 |
| July | 257 | 607 | 726 | 643 |
| August | 35 | 250 | 596 | 313 |
| September | 0 | 538 | 718 | 217 |
| October | 0 | 714 | 212 | 609 |
| November | 0 | 653 | 588 | 717 |

Table 1:  Number of points used for median calculation.

| T-test results | AFR |
|---|---|
| AM | 1.128E-14 |
| AAC | 2.2E-16 |
| NAA | 0.02458 |
| NCA | 0.003387 |
| EUR | 2.69E-12 |
| EUR/AFR | 0.03947 |

Table 2: Results of T-test with Afr.

| Airmass Season | AM | AAC | NAA | NCA | EUR | AFR | EUR/AFR |
|---|---|---|---|---|---|---|---|
| Winter 11 | 0 | 8 (1.26 +/- 0.01) | 4 (1.32 +/- 0.01) | 6 (1.36+/- 0.04) | 59 (1.33 +/- 0.06) | 350 (1.21 +/- 0.12) | 184 (1.3 +/- 0.07) |
| Spring 12 | 69 (1.23 +/- 0.035 | 142 (1.26 +/- 0.05) | 77 (1.28 +/- 0.035) | 138 (1.31 +/- 0.063) | 83 (1.29 +/- 0.06) | 110 (1.26 +/- 0.07) | 33 (1.26 +/-0.11) |
| Summer 12 | 11 (1.19 +/- 0.03) | 66 (1.21 +/- 0.04) | 0 | 6 (1.27 +/-0.04) | 14 (1.21 +/- 0.04) | 0 | 0 |
| Autumn 12 | 0 | 0 | 0 | 0 | 0 | 0 | 0 |
| Winter 12 | 0 | 19 (1.17 +/- 0.05) | 13 (1.23 +/- 0.09) | 54 (1.24 +/- 0.05) | 35 (1.17 +/-0.04) | 120 (1.19 +/- 0.09) | 3 (1.21 +/-0.07) |
| Spring 13 | 67 (1.20 +/- 0.06) | 95 (1.19 +/- 0.09) | 86 (1.20+/- 0.06) | 109 (1.21 +/- 0.06) | 28 (1.13 +/- 0.1) | 45 (1.19 +/- 0.08) | 0 |
| Summer 13 | 218 (1.25 +/- 0.05) | 10 (1.26 +/- 0.09) | 17 (1.12+/- 0.09) | 58 (1.24 +/- 0.06) | 217 (1.25 +/- 0.1) | 0 | 4 (1.31 +/- 0.03) |
| Autumn 13 | 53 (1.13 +/- 0.14) | 205 (1.14 +/- 0.08) | 9 (1.16 +/- 0.05) | 24 (1.09 +/- 0.07) | 121 (1.2 +/- 0.08) | 106 (1.17+/- 0.11) | 118 (1.17 +/-0.06) |
| Winter 13 | 41 (1.17 +/- 0.06) | 22 (1.19 +/- 0.06) | 51 (1.27 +/- 0.03) | 186 (1.25 +/-0.06) | 13 (1.33 +/- 0.13) | 328 (1.29 +/- 0.2) | 56 (1.31 +/-1.27) |
| Spring 14 | 37 (1.2 +/- 0.04) | 51 (1.17 +/- 0.04) | 81 (1.22 +/- 0.04) | 75 (1.22 +/- 0.04) | 39 (1.21 +/- 0.04) | 32 (1.21 +/- 0.05) | 57 (1.22 +/- 0.05) |
| Summer 14 | 0 | 192 (1.15 +/- 0.05) | 10 (1.18 +/- 0.03) | 36 (1.16 +/- 0.05) | 91 (1.17 +/-0.07) | 0 | 0 |
| Autumn 14 | 20 (1.10 +/- 0.15) | 191 (1.12 +/-0.05) | 37 (1.21 +/- 0.04) | 102 (1.17 +/- 0.05) | 42 (1.14 +/- 0.06) | 80 (1.15 +/- 0.06) | 36 (1.2 +/- 0.08) |
| Winter 14 | 26 (1.11 +/- 0.09) | 79 (1.22 +/- 0.16) | 16 (1.13 +/- 0.04) | 24 (1.07 +/- 0.07) | 126 (1.09 +/- 0.1) | 15 (1.06+/- 0.04) | 25 (1.05 +/- 0.03) |
| Spring 15 | 38 (1.03 +/- 0.05) | 95 (1.04 +/- 0.08) | 20 (1.04 +/- 0.02) | 76 (1.03 +/- 0.09) | 174 (1.02 +/- 0.14) | 3 (1.05 +/- 0.04) | 32 (1.04 +/- 0.03) |
| Summer 15 | 48 (0.967 +/- 0.14) | 280 (0.983 +/- 0.1) | 14 (1.03 +/- 0.04) | 49 (1.01 +/- 0.05) | 96 (1.03 +/- 0.11) | 2 (1.25 +/-0.03) | 0 |
| Autumn 15 | 15 (1.18 +/- 0.08) | 84 (1.16 +/-0.11) | 15 (1.21 +/- 0.03) | 37 (1.16 +/- 0.06) | 56 (1.16 +/- 0.06) | 269 (1.19+/- 0.16) | 39 (1.18 +/- 0.05) |

Table 3: The number of points that have been used for the median calculation used in Figure 10. The median and standard deviation is included in brackets (ng m$^{-3}$).

a)

[Figure]

b)

[Figure]

Figure 3a) 10-day back trajectories for the measurement period arriving at ground level to the CVAO run using Hysplit (Carslaw et al., 2012), b) All 10-day back trajectories to the CVAO arriving at 1.5 km.

Carslaw, D.C. and K. Ropkins, (2012). openair — an R package for air quality data analysis. *Environmental Modelling & Software*. Volume 27-28, 52-61.